# Rice *OsHSFA3* Gene Improves Drought Tolerance by Modulating Polyamine Biosynthesis Depending on Abscisic Acid and ROS Levels

**DOI:** 10.3390/ijms21051857

**Published:** 2020-03-09

**Authors:** Ming-Dong Zhu, Meng Zhang, Du-Juan Gao, Kun Zhou, Shan-Jun Tang, Bin Zhou, Yan-Mei Lv

**Affiliations:** 1Hunan Rice Research Institute, Key Laboratory of Indica Rice Genetics and Breeding in the Middle and Lower Reaches of Yangtze River Valley, Ministry of Agriculture, Changsha 410125, China; uhz_uhz@hotmail.com (M.-D.Z.); gaodujuan221@163.com (D.-J.G.); zhzhkp@hotmail.com (K.Z.); shanjunt@126.com (S.-J.T.); 2Hunan Key Laboratory of Plant Functional Genomics and Developmental Regulation, College of Biology, Hunan University, Changsha 410082, China; zhangmeng2019@hnu.edu.cn

**Keywords:** drought, abscisic acid, gene expression, polyamines, reactive oxygen species

## Abstract

Drought is a serious problem, which causes heavy yield losses for rice. Heat-shock factors (HSFs) had been implicated in tolerance to drought and high temperature. However, there has not been much functional characterization and mechanism clarification in rice. Previously, we found an HSF gene, *OsHSFA3*, was highly related with drought tolerance after screening from 10,000 different samples. Herein, we cloned the *OsHSFA3* from rice and overexpressed it in *Arabidopsis thaliana* to study its regulatory mechanism of drought tolerance. Phenotypic and physiological assays of the transgenic *Arabidopsis* lines showed that overexpression of *OsHSFA3* confers drought tolerance by reducing water loss and reactive oxygen species (ROS) levels, whereas it increases abscisic acid (ABA) levels. However, enzymatic antioxidants such as activity levels of superoxide dismutase, peroxidase and catalase were not significantly different between wild type and transgenic lines. Instead, we observed a significant increase in polyamine content, which was correlated with increased *AtADC1, AtADC2, SPDS1* and *SPMS* expression levels. In silico and in vivo analyses confirmed that *OsHSFA3* is a nuclear-localized gene. In addition, *OsHSFA3* can bind to the promoter of *AtADC1* and *OsADC* via a yeast one-hybrid assay. Overall, this study reveals that *OsHSFA3* improves drought tolerance in *Arabidopsis* not only by increasing ABA levels, but also by modulating polyamine levels to maintain ROS homeostasis, therefore it could be a strong candidate to develop drought-tolerant rice cultivars.

## 1. Introduction

Abiotic stresses such as drought, salinity and suboptimal temperature are serious threats for agriculture and global food security [1,2]. These stresses negatively affect plant growth and development and ultimately result in poor crop production [3,4,5]. Rice is a major grain crop and feeds about half of the world population [6,7]. Drought is one of the major abiotic stresses negatively influencing rice grain yield [8,9]. It causes osmotic stress to plants and leads to over-accumulation of reactive oxygen species (ROS) in different cellular compartments [10]. This over-accumulation of ROS causes oxidative stress to plants and damages various cellular organelles including the cell membrane [11,12]. It also leads to early senescence of leaves and other tissues via enhanced programmed cell death [11,13].

Since plants cannot move from one place to other to avoid the harmful effects of stresses, they adopt various mechanisms for stress tolerance. The antioxidant defense system is one of the major tolerance mechanisms for ROS scavenging under abiotic stresses [14]. Enzymatic antioxidants including superoxide dismutase (SOD), peroxidase (POD) and catalase (CAT) are key ROS-scavenging enzymes [15,16]. Similarly, polyamines are vital plant metabolites that serve as important ROS scavengers and also induce stomatal closure under drought stress [17,18]. The enzyme arginine decarboxylase (ADC) catalyzes arginine to agmatine, which is the precursor of putrescine and higher polyamines (PAs), such as spermidine and spermine. Several genes have been reported to control the production of polyamines in plants [18,19]. In *Arabidopsis thaliana*, ADC only has two paralogues, *AtADC1* and *AtADC2* [20]. In rice, several genes regulate the biosynthesis of polyamines in a collective way and contribute to drought tolerance [21,22]. Thus, exploiting such tolerance mechanisms provide an amazing option to cope with drought stress in rice.

Transcription factors (TFs) are well-known for their role in abiotic stress tolerance in plants [8,23]. Among the different TF families, heat shock factors (HSFs), WRKY-motif containing proteins (WRKY), myeloblastosis (MYB) and APETALA2/ethylene response factor (AP2/ERF) families are the most promising for their role in drought and other abiotic stress tolerance [18,24,25,26,27]. HSF is a large gene family in rice with 25 members and several members of this family have been characterized for their role in stress tolerance [28,29]. Abscisic acid (ABA) and ROS are important signaling molecules and regulate the expression of various stress responsive genes [16,30,31]. In rice, expression levels of HSF genes *OsHsfA4a* and *OsHsfA2* were upregulated under ABA and ROS treatments and thus regulate stress tolerance [32]. Overexpression of rice *OsHsfC1b* in *Arabidopsis* causes salinity tolerance in transgenic plants [33]. Similarly, overexpression of rice *OsHsfA2e* in *Arabidopsis* enhanced tolerance against multiple abiotic stresses [34].

Previously, we observed drought-resistant lines from more than 10,000 rice cultivars, and finally we found that *OsHSFA3* might be the core drought-responsive gene in a drought-tolerant cultivar ‘Hunan’ [35]. To further evaluate the function of *OsHSFA3* and clarify its mechanism in drought tolerance, we applied the model plant *Arabidopsis thaliana* for better understanding. Finally, our results demonstrated that *OsHSFA3* imparts drought tolerance by regulating ROS and polyamine biosynthesis levels.

## 2. Results

### 2.1. OsHSFA3 Is a Drought-Responsive Gene and Highly Accumulated in Shoot

Previous transcriptome analysis, gene coexpression analysis and genome-wide studies have identified a wide range of heat shock factors (HSFs) in the rice genome [28,36]. Of these rice HSFs, *OsHSFA3* was reported to be responsive to cold and drought stress [35,36]. The homologous gene of *OsHSFA3* in *Arabidopsis thaliana* (AT5G03720, AtHSFA3) is quite different in terms of gene structure (two introns and three exons) and protein similarity (~40% identity), indicating that OsHSFA3 and AtHSFA3 may have distinct functional properties. However, its functional characterization has not been reported yet. Here we performed a detailed functional characterization of *OsHSFA3* to understand its molecular mechanism for drought tolerance.

To confirm if *OsHSFA3* is responsive to drought stress, we measured mRNA abundance of *OsHSFA3* under different drought stress durations using real-time quantitative PCR (qRT-PCR). *OsHSFA3* mRNA accumulated in rice within two days of drought treatment and increased many fold in a time-dependent manner (Figure 1A). After four days of drought stress, the relative expression showed a two-fold increase as compared to the relative expression after two days under stress. Further extension of drought stress duration to seven days resulted in higher relative expression, indicating that *OsHSFA3* is a drought-responsive gene (Figure 1A). We then examined the tissue-specific expression pattern of *OsHSFA3* in four rice tissues, i.e., leaf, shoot, root and seed. *OsHSFA3* was expressed in all four organs, however, the highest relative expression was observed in shoot followed by leaf, root and seed (Figure 1B). The relative expression levels in root and seed were relatively lower than in the other organs.

### 2.2. Structural Features of OsHSFA3 and Evolutionary Relationship

The *OsHSFA3* gene sequence was obtained from the Rice genome annotation project (http://rice.plantbiology.msu.edu/) under gene locus *LOC_Os02g32590*. The *OsHSFA3* gene located on chromosome 2 of rice has a full-length CDS (coding sequence) of 1497 bp which encodes a polypeptide of 498 amino acids, consisting of one HSF domain (64–157 aa), one low-complexity domain (15–21 aa), a coiled-coil domain (180–227 aa), and a low-complexity domain (345–357 aa) (Appendix A). Gene structure analysis indicated that the *OsHSFA3* gene has one intron of 1298 bp and two exons (Appendix A). To investigate the evolutionary relationship of *OsHSFA3* with its homologs from different plant species, a phylogenetic tree was constructed using orthologues of OsHSFA3 proteins from 13 plant species. The phylogenetic analysis divided the OsHSFA3 orthologues into two main groups, that is, monocots and dicots. Logically, OsHSFA3 falls into the group of monocots suggesting that it is closely related to the members of HSFs in monocots; particularly grouped with HSFs of *Setariaitalica*, *Zea mays* and *Sorghum bicolor* (Figure 1C). We then performed alignment of these protein sequences using ClustalW to identify the conserved regions among different species. The alignment demonstrated that regions containing an HSF domain (64–157 aa), coiled-coil domain (180–227 aa), and a low-complexity domain (345–357 aa) were highly conserved among all species (Figure 1D). However, the other protein sequence is mostly conserved among the monocots, while considerable variation is present with dicots (Figure 1D).

### 2.3. OsHSFA3 Localized in Nucleus

To see the subcellular localization of *OsHSFA3*, we first predicted its localization using bioinformatics tools. The online tool WoLF PSORT predicted its subcellular localization in the nucleus. To verify the subcellular localization, the full-length *OsHSFA3* CDS without the termination codon was fused with the green fluorescence protein (GFP) coding sequence in vector pBWA(V)HS-GLosGFP to express the OsHSFA3-GFP recombinant protein. The constructed plasmid was then transformed into *Arabidopsis* protoplasts and observed under a confocal laser microscope. The GFP signals were colocalized with nuclear marker signals and merged well, which confirmed that *OsHSPA3* was localized in the nucleus (Figure 2A–E). We also transformed a control plasmid containing only the GFP sequence without *OsHSFA3* into *Arabidopsis* protoplasts. As expected, this construct displayed GFP signals in all parts of cell, not only in nucleus, proving the reliability of our results (Figure 2F–I).

### 2.4. Overexpression of OsHSFA3 Increases Osmotic and Drought Stress Tolerance in Transgenic Arabidopsis

Given that *OsHSFA3* showed a strong induction under drought stress, we speculated that *OsHSFA3* may play an important role in drought stress response. To test our hypothesis, we overexpressed *OsHSFA3* in *Arabidopsis* using CaMV 35S promoter. The overexpression vector was transformed into *Arabidopsis* ecotype Col-0 cv. by the floral dip method using *Agrobacterium tumefaciens* strain LBA4404. Of the total 11 positive transgenic lines, three independent lines expressing the transgene (designated as L1, L2 and L3 hereafter) were selected for further experiments along with the wild type (WT), i.e., Col-0 (Appendix A).

We first imposed osmotic stresses to WT and transgenic *Arabidopsis* plants by supplementing 150 mM mannitol to 10-day-old seedlings for 10 days and analyzed the root lengths (Figure 3A). Overall, the root growth was reduced in the presence of mannitol. However, significantly longer root lengths were observed in transgenic lines as compared to WT seedlings (Figure 3B). We also treated 10-day-old seedlings with 1 μM ABA and observed that root length reduced significantly in ABA-treated seedlings as compared to the seedlings treated as control. An important observation was the significant reduction of root length in all transgenic lines as compared to WT when treated with ABA (Figure 3B). These findings suggested that *OsHSFA3* is responsive to osmotic stress and ABA.

Following osmotic stress, one-month-old plants were submitted to drought stress for 20 days by terminating watering (Figure 4A). Thereafter, seedlings were transferred to normal growth conditions (watering was resumed) and grown for five days. The survival rate of WT plants was 26%, while an average of 83.3% of transgenic plants recovered (Figure 4B). Similarly, a decrease in biomass was observed under drought stress in all plants. However, the transgenic lines showed significantly higher biomass as compared to WT under drought stress (Figure 4C). Fresh weight is an important indicator of stress tolerance in plants. There was no significant difference in fresh weight among WT and transgenic plants under control conditions (Figure 4D). Nevertheless, a significantly higher fresh weight (FW) was observed in all transgenic lines as compared to WT under drought stress (Figure 4D). Water loss rates of detached leaves have been widely used to reflect drought tolerance in plants [18]. Water loss rates were assessed after every 30 min for both transgenic plants and control plants. We witnessed a steady leaf water loss with increasing time after dehydration in all plants. Notably, the water loss rate was higher in WT as compared to overexpressing transgenic lines (Figure 4E). These findings demonstrate that the transgenic lines were more resistant to dehydration and were drought stress tolerant.

### 2.5. Overexpression of OsHSFA3 Increases Drought Tolerance by Decreasing H_2_O_2_ Accumulation and Increasing ABA Levels

ROS accumulation is a common response under abiotic stresses which causes oxidative stress to plants [14]. Considering its vital role in drought tolerance, we investigated H_2_O_2_ levels in the WT and *OsHSFA3-*overexpressing lines. After 20 days under drought stress conditions, we witnessed an increase in H_2_O_2_ levels in WT and transgenic lines as compared to control conditions (Figure 5A). Significantly higher H_2_O_2_ levels were recorded in WT plants relative to transgenic plants (Figure 5A). Among transgenic lines, L2 and L3 had higher levels while L1 had slightly lower H_2_O_2_ levels. These results showed that WT plants accumulated more H_2_O_2_ as compared to transgenic lines under drought stress and thus were more sensitive to oxidative stress. Membrane lipid peroxidation is an important stress biomarker under different abiotic stresses. H_2_O_2_ accumulation normally enhances lipid peroxidation as a result of oxidative damage to cells [37]. Membrane lipid peroxidation is measured in terms of malondialdehyde (MDA). In this study, we observed a significant increase in MDA levels under drought stress in the WT as compared to transgenic plants (Figure 5B). These results suggested that *OsHSFA3-*overexpressing plants had maintained ROS homeostasis and thus had lower oxidative damage.

Enzymatic antioxidants play important roles in ROS scavenging for normal plant growth [13,38]. To estimate the ability of *OsHSFA3-*overexpressing *Arabidopsis* lines and WT to scavenge ROS, the activities of antioxidant enzymes, i.e., SOD, POD and CAT, were measured. The activities of the three antioxidants were not apparently different between transgenic lines and WT plants under normal conditions. Observably, the response of transgenic and WT plants under drought was higher in terms of enzyme activities. However, no obvious differences were observed between WT and transgenic lines. These results suggest the overexpression of *OsHSFA3* did not significantly modify the scavenging ability of transgenic *Arabidopsis* plants as compared to WT (Figure 5C–E).

ABA is an important plant stress signaling hormone which plays important role in drought stress tolerance via regulating expression of various stress-responsive genes [31,39,40]. Thus, we measured endogenous ABA levels in *OsHSFA3-*overexpressing lines and WT plants. ABA levels were significantly higher in drought-stressed plants as compared to control plants (Figure 5F). ABA content was higher in all transgenic lines than in WT plants after they were subjected to drought conditions. Under normal conditions, the ABA levels were very low and were not significantly different between WT and transgenic lines (Figure 5F). These results suggest that *OsHSFA3* may play an important role in ABA signaling. These findings also correlate with the results of exogenous application of ABA to transgenic and WT seedlings (Figure 3B).

### 2.6. OsHSFA3 Regulates Polyamines Biosynthesis under Drought Stress

Further work was carried out to elucidate the molecular mechanism underlying the enhanced drought tolerance rendered by *OsHSFA3*. Commonly, plants would modulate a battery of functional genes and TFs, including HSFs, which are involved in the synthesis of various defense metabolites to resist the abiotic stress. To validate whether HSFs were involved in defense metabolite synthesis, we determined the accumulation of polyamines, i.e., diamine putrescine (Put), triamine spermidine (Spd) and tetraminespermine (Spm) [18]. We attempted to analyze whether polyamine synthesis was altered in the transgenic plants. In response to drought stress, Put levels were significantly increased in transgenic plants as compared to WT. Under controlled conditions, the levels were lower but still a difference between WT and transgenic lines existed (Figure 6A). Regarding Spd, we observed an increasing trend under drought stress as compared to control and a significant difference between WT and transgenic lines under both control and drought stress conditions was observed (Figure 6B). Notably, Spd levels were significantly higher in overexpressing lines as compared to WT, which suggested its putative role in drought tolerance. In the case of Spm, drought treatment led towards no significant changes in levels of Spm (Figure 6C). However, transgenic overexpressing lines still had significantly higher Spm levels as compared to WT (Figure 6C). These results concluded that *OsHSFA3* regulates polyamines biosynthesis particularly under drought stress conditions, and that polyamines may play important roles in drought tolerance of the transgenic *Arabidopsis* plants overexpressing *OsHSFA3*.

### 2.7. Expression of Polyamine Biosynthesis Genes is Induced under Drought Stress

The key enzymes which regulate the polyamine biosynthesis include arginine decarboxylase (ADC), spermidine synthase (SPD) and spermine synthase(SPMS) [41]. In *Arabidopsis thaliana*, ADC production is regulated by two ADC paralogues, *AtADC1* and *AtADC2* [20]. Similarly, regulation of SPD and SPMS is controlled by the expression of the *SPDS1* and *SPMS* genes, respectively [41]. We hypothesized that *OsHSFA3* may regulate the expression of these key genes for drought tolerance. Thus, we analyzed the mRNA abundance levels of these genes under drought stress treatment in WT and *OsHSFA3*-overexpressing lines. The results showed that the expression of all tested genes was upregulated under drought conditions in both WT and transgenic plants, denoting that these genes are highly drought-responsive (Figure 6D–G). Notably, the mRNA abundance of the *AtADC1* gene was significantly higher under drought in transgenic plants as compared to WT, indicating that *AtADC1* could be directly influenced by *OsHSFA3*.

### 2.8. OsHSFA3 Interacts with OsADC1 and AtADC1 Promoter

HSF transcription factors normally regulate the expression of various stress-responsive genes via binding to heat shock elements (HSEs) [23]. In silico cis-element analysis of the promoter of *OsADC1* and *AtADC1* showed that they contain HSE elements, so can be effectively bound by HSFs. To determine if *OsHSFA3* is involved in the activation of drought stress signaling pathways, we performed a yeast one-hybrid assay. The interaction between prey and bait was observed according to the growth of the yeast transformants in a series of 10-fold dilutions. The results showed that the full length *OsHSFA3* could significantly bind to *OsADC1* (*LOC_Os08g33620*) as well as to the *AtADC1* promoter (Figure 6H). These results indicate that *OsHSFA3* was able to activate the expression of *OsADC1* and *AtADC1* by binding related cis-elements.

## 3. Discussion

Heat shock transcription factors (HSFs) constitute a large TF family with evident roles in growth, development and reproduction [36,42]. In addition, their important role in abiotic stress tolerance has emerged recently in several crop species [23,24,34,43]. HSFs are comprised of 25 members in rice and 22 members in *Arabidopsis* [28]. Although several members of rice HSFs have been reported for their role in growth and stress responses, the function and mechanism of action of many HSF genes are yet to be studied. The rice HSF TF *OsHSFA3* has been shown to be responsive to cold and drought stress in rice [35,36]. However, how *OsHSFA3* regulates drought tolerance in rice is not known yet. In this study, we confirmed that *OsHSFA3* is induced upon drought treatment and the transcript level continued to increase with the duration of stress level, suggesting that *OsHSFA3* may regulate drought tolerance in rice (Figure 1). Phylogenetic analysis with the orthologue genes indicated a close similarity with other monocots, particularly *Setariaitalica*, *Zea mays* and *Sorghum bicolor.* To further see the typical feature of a TF, we observed its subcellular localization in *Arabidopsis* protoplasts using GFP-fused protein. *OsHSFA3* has shown nuclear localization, which was in accordance with other HSF TFs [34,44].

*Arabidopsis* serves as an excellent model plant to study the function of important genes belonging to different crop species, especially for abiotic stresses [31,34,41,45,46]. We thus overexpressed *OsHSFA3* in *Arabidopsis thaliana* to study its response to drought stress. Root is an important agronomic trait that helps plants survive under limiting water conditions. Plants adjust their root length under drought conditions to cope with the water deficiency. We observed significant increases in the root length under mannitol treatment in transgenic *Arabidopsis* plants overexpressing *OsHSFA3* as compared to WT (Figure 3), suggesting that *OsHSFA3* may positively regulate drought tolerance. ABA is an important phytohormone that acts as a signaling molecule to activate the expression of various TFs under abiotic stress [47,48,49]. We observed a significant decrease in root length under ABA treatment in *OsHSFA3-*overexpressing plants as compared to WT, suggesting that *OsHSFA3* is hypersensitive to ABA and ABA may regulate the expression of *OsHSFA3* (Figure 3). Consistently, we observed higher ABA levels under drought stress in *OsHSFA3-*overexpressing plants as compared to WT (Figure 5), denoting that *OsHSFA3* may work in the ABA-dependent pathway.

The most important feature showing the drought tolerance level in plants is the survival ability under drought stress [47,50,51]. We observed a significantly higher survival ability under drought in the *OsHSFA3-*overexpressing plants as compared to WT, suggesting an important role of *OsHSFA3* in drought tolerance. Similarly, *OsHSFA3-*overexpressing plants maintained higher biomass as compared to WT under drought stress, which further revealed its role in drought tolerance. These observations were in line with previous findings which revealed higher plant biomass and survival ability in drought-tolerant plants [40,52,53]. These findings indicate that *OsHSFA3* induces drought tolerance in transgenic plants by different plant adaptations such as longer root length, reduced water loss and modulating the ABA level to protect from dehydration.

Under different abiotic stresses including drought, the production of ROS is increased many fold, which leads to oxidative stress in plants [1,30]. Higher ROS levels induce lipid peroxidation in plants and cause injury to cell membranes [10,15]. In this study, we observed a sharp increase in H_2_O_2_ and MDA levels under drought stress in WT plants, but H_2_O_2_ and MDA levels were significantly less in *OsHSFA3-*overexpressing plants (Figure 5). This indicates a putative role of *OsHSFA3* in ROS scavenging and drought tolerance. Plants normally protect themselves from oxidative stress by activating their antioxidant defense systems [14,30]. SOD, CAT and POD are the most common antioxidant enzymes which scavenge ROS by detoxifying H_2_O_2_ and O_2_^−^ into water and stable oxygen [30]. Thus, these antioxidant enzymes play a key role in stress tolerance by ROS scavenging [1,54]. Although we observed a sharp increase in the activities of these antioxidant enzymes under drought stress, there was no significant difference in their activities among WT and *OsHSFA3-*overexpressing plants (Figure 5). This suggests that *OsHSFA3* does not regulate the activities of antioxidant enzymes, and that *OsHSFA3* plays a role in ROS scavenging by some other mechanism [16]. Further, we investigated only three key antioxidant enzymes (SOD, POD and CAT), so the possibility that other antioxidants not studied here were activated could not excluded.

In plants, PAs were found to play crucial roles not only in plant growth, physiology and reproduction, but also in conferring abiotic stress (salt, drought and metal/metalloid toxicity) tolerance by enhancing antioxidant defense and oxidative stress tolerance [21,41,55]. In this study, we observed a significant increase of PA content in *OsHSFA3-*overexpressing plants than in WT under both control as well as drought stress conditions (Figure 6). This increased PA level was correlated with increased expression of *AtADC1, AtADC2, SPDS1* and *SPMS* genes, which are the key genes regulating the PA level [19,20,21]. In addition, we showed that *OsHSFA3* can bind to *AtADC1* and *OsADC* via a yeast one-hybrid assay (Figure 6). Taken together, these findings demonstrated that *OsHSFA3* improves drought tolerance in *Arabidopsis* by modulating polyamine levels to maintain ROS homeostasis.

## 4. Materials and Methods

### 4.1. Plant Growth Conditions and Stress Treatments

Rice seedlings were grown in soil mixed with compost in a growth chamber at 28 °C with 16 h light and 8 h dark. Drought stress was imposed to rice seedlings by withholding water for 7 days on 3-week-old plants. Control plants were regularly watered to ensure normal plant growth. *Arabidopsis thaliana* seeds were grown in 340 mL pots filled with a mixture of peat/forest soil and vermiculite (3:1) in a glasshouse at 21 °C with a light intensity of 50 μmol·m^−2^·s^−1^ and 70% relative humidity under 16 h light/8 h dark conditions. Then, these seedlings were transferred into MS medium supplemented with 3% (*w*/*v*) sucrose and 0.8% agar and were grown under the same growth conditions with *Arabidopsis* seeds. The plate-grown seedlings were transferred to soil after 1 week. For drought stress, 4-week-old plants were subjected to 20 days of drought stress by withholding water supply. Control (Ck) plants were regularly watered to ensure normal growth.

### 4.2. Total RNA Isolation and qRT-PCR

Total RNA from untreated and drought-treated plants was isolated using the RNA extraction kit (Tiangen, Beijing, China), then the first-strand cDNA was synthesized from 500 ng total RNA by reverse transcriptase (Invitrogen, Carlsbad, CA, USA). The cDNA was diluted ten times to use as template in qRT-PCR with SYBR Premix ExTaq Mix (Takara, Dalian, China). The procedures were followed as described earlier [35]. Primer sequences are mentioned in Appendix A. The housekeeping actin gene was used as the reference gene.

### 4.3. Gene structure Prediction and Phylogenetic Analysis

Gene structure, including introns and exons of the *OsHSFA3* gene, was investigated by using the online Gene Structure Display Server (http://gsds.cbi.pku.edu.cn/) as described earlier [23]. The genomic and coding sequences were retrieved from rice genome annotation project (http://rice.plantbiology.msu.edu/).

Homologue alignment was obtained using the online tool ExPASy bioinformatics resource portal (https://embnet.vital-it.ch/software/BOX_form.html). Amino acid sequences of all orthologs were retrieved from NCBI. Sequences were aligned using ClustalW and a phylogenetic tree was constructed using MEGA 7 software (Temple University, Philadelphia, PA, USA) using neighbor-joining method with 1000 bootstrap replicates.

### 4.4. Vector Construction and Transformation

To investigate the tolerance to drought and osmotic stress of *OsHSFA3* transgenic lines after molecular identification, the vector overexpressing pCAMBIA 1301s (which is a modified form of the pCAMBIA1301 vector) was generated by inserting *OsHSFA3* into *Kpn* I (5′-end) and *Bam*H I (3′-end) sites, driven by the CaMV 35S promoter (Appendix A). The overexpressing vectors were then transformed to *Arabidopsis* ecotype col-0 cv by the floral dip method using *Agrobacterium tumefaciens* strain LBA4404. Eleven positive independent transgenic lines were obtained and three independent overexpression lines of *Arabidopsis* with high transcript levels of *OsHSFA3* were selected for further experiments along with the control, i.e., Col-0.

### 4.5. Subcellular Localization

The subcellular localization was first predicted using the online bioinformatic tool, WoLF PSORT (www.genscript.com/tools/wolf-psort). For experimental validation, the *OsHSFA3* coding sequence was fused with the GFP coding sequence into a pBWA(V)HS-GLosGFP vector and the nucleus marker vector (pBWA(V)HS-Nucleus-mKate) was cotransformed into *Arabidopsis* protoplasts. The transient expression of OsHSFA3-GFP was observed in *Arabidopsis* protoplasts under a confocal laser microscope. For the negative control, the empty vector containing only the *GFP* gene was also transformed into *Arabidopsis* protoplasts.

### 4.6. Measurement of ROS, MDA and Antioxidant Activities

ROS were measured quantitatively in terms of H_2_O_2_ as described earlier [13]. Briefly, 100 mg of fresh tissue was harvested and grounded with liquid nitrogen. The fine powder was immersed with 1 mL buffer of 50 mM sodium phosphate (pH 7.4) for 20 min on ice. After centrifuging for 15 min at 12,000× *g* and 4 °C, H_2_O_2_ in the supernatant was quantified using spectrophotometer. H_2_O_2_, MDA, SOD, POD and CAT were assayed by using specific kits, respectively (Nanjing Jiancheng bioengineering Institute, Nanjing, China).

### 4.7. Analysis of Polyamines and ABA

PAs were assayed by high-performance liquid chromatography (HPLC) using the protocol described before [56]. PA showed a maximum peak at 254 nm by detecting with a UV detector. Soluble conjugated PA were calculated by subtracting the free PA from the acid-soluble PA. ABA was measured using the gas chromatography/mass spectrometry (GC-MS) method as described by Okamoto et al. [57].

### 4.8. Yeast One Hybrid Assay

Putative HSE cis-elements were identified in the promoter region of *OsADC* (*LOC_Os08g33620*) and *AtADC1*(*At2g16500*). Yeast one-hybrid assay was performed to investigate whether OsHSFA3 could bind with these HSE cis-elements in the promoters of *OsADC* and *AtADC1*. The full-length ORF of *OsHSFA3* was amplified by PCR using gene-specific primers (Appendix A) and cloned into pGADT7-Rec (Clontech, Mountain View, CA, USA) to create a prey vector (pGADT7-OsHSFA3). About 2000 bp promoter sequences of *OsADC* and *AtADC1* genes were respectively cloned into the bait vector. The *OsHSFA3* prey vector was cotransformed with either the *OsADC* or *AtADC1* bait vector into yeast strain Y187. The cotransformed yeasts containing the bait and prey were cultivated on the SD/-Leu/-Trp/-His selective media supplemented with 0 or 3 mM 3-amino-1,2,4-triazole (3-AT) for 3 days according to the instructions for the Matchmaker^TM^ Gold Yeast One Hybrid System (Clontech, USA). Yeasts cotransformed with pGADT7-Rec2-53 (the pGADT7-Rec2-53 plasmid containing a murine *p53* fused to *GAL4 AD* domain) and p53HIS2 (the p53HIS2 harboring three tandem copies of p53-binding DNA elements) were used as the positive control. The negative control was pGADT7-Rec2-OsHSFA3 and p53HIS2 cotransformation. The interaction between prey and bait was observed according to the growth of the yeast transformants in a series of 10-fold dilutions.

## 5. Conclusions

In summary, this study revealed the role of rice *OsHSFA3* as a positive regulator of drought tolerance [2]. Collectively, the results indicate that *OsHSFA3* works in an ABA-dependent pathway to limit water loss and maintains ROS homeostasis via regulating PA levels and the expression of various PA biosynthesis genes (Appendix A). Further studies could unravel more downstream targets of *OsHSFA3* to have a deeper understanding of the drought signaling network

## Figures and Tables

**Figure 1 ijms-21-01857-f001:**
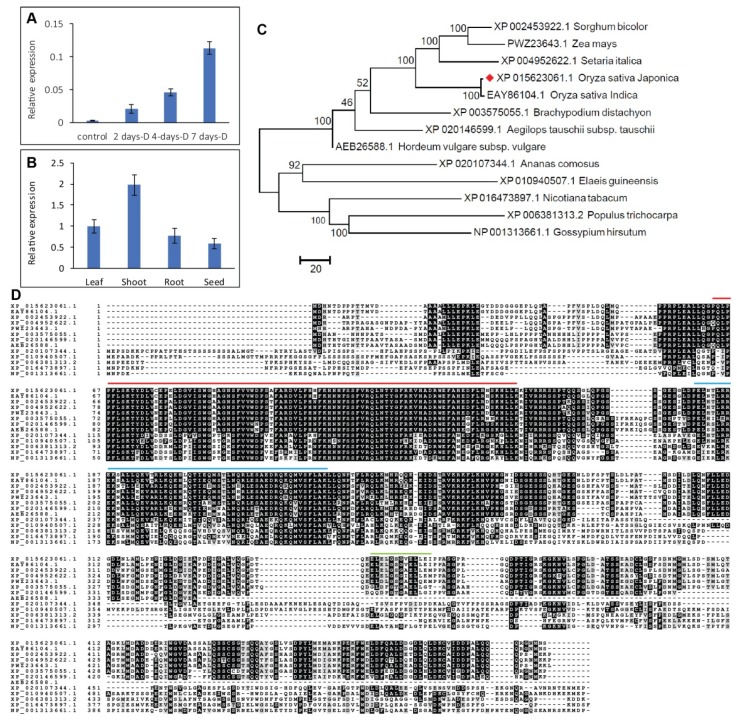
Expression pattern and sequence analysis of *OsHSFA3*. (**A**) Relative expression of *OsHSFA3* under control and time course drought treatment in rice. (**B**) Relative expression of *OsHSFA3* in different rice tissues. (**C**) Maximum likelihood phylogenetic tree based on amino acid sequences of *OsHSFA3* and orthologs from different species. Red rhombus is the OsHSF3. (**D**) Multiple sequence alignment showing heat-shock factor (HSF) domain (64–157 aa, under red bar), coiled-coil domain (180–227 aa, under blue bar), and a low-complexity domain (345–357 aa, under green bar).

**Figure 2 ijms-21-01857-f002:**
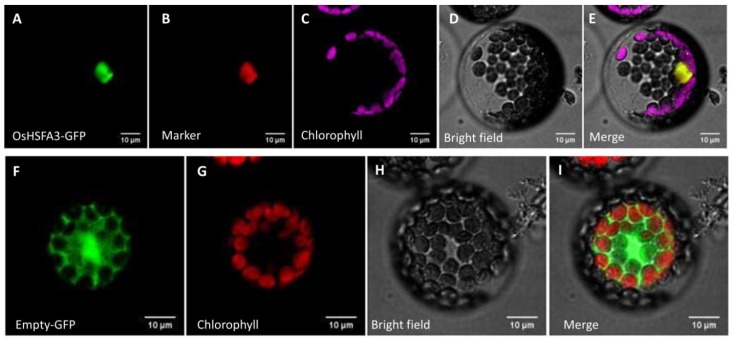
Subcellular localization of OsHSFA3 protein. *Arabidopsis* protoplasts were transiently transformed with constructs containing either OsHSFA3 fused with green fluorescent protein (GFP) (**A**–**E**) or GFP alone (**F**–**I**).

**Figure 3 ijms-21-01857-f003:**
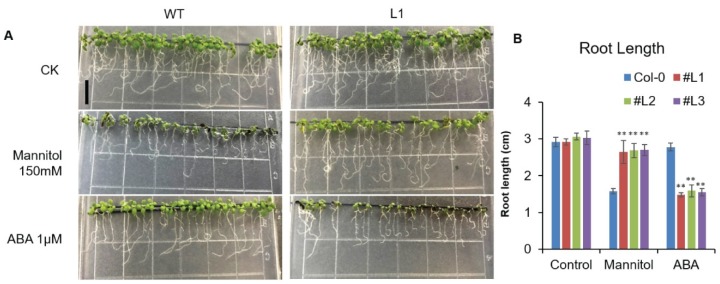
Root length observation of wild type (WT) and *OsHSFA3*-overexpressing transgenic *Arabidopsis* plants with mannitol and abscisic acid (ABA) treatment. (**A**) Seedlings of wild type and *OsHSFA3* transgenic lines grew on MS medium in control (CK), mannitol treatment (150 mM) and ABA treatment (1 µM). (**B**) Statistical analysis of root length of wild type and *OsHSFA3*-overexpressing *Arabidopsis* plants with and without mannitol and ABA treatment. Values indicate mean ± SE of 10 biologically independent samples. Significance of data is tested with Student’s *t* test. ** represents *p* < 0.01.

**Figure 4 ijms-21-01857-f004:**
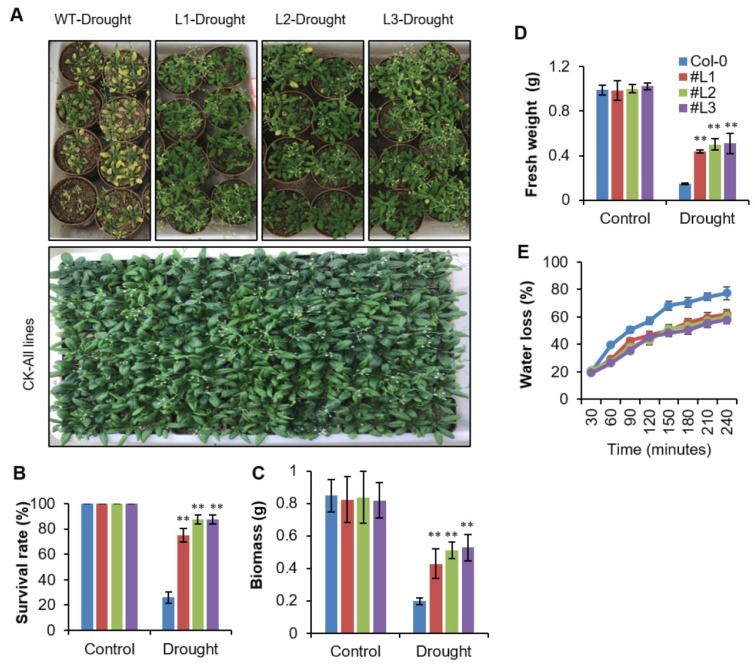
Assay of drought tolerance and associated traits for *OsHSFA3* overexpression transgenic *Arabidopsis* plants. (**A**) Phenotypic analysis of one-month old WT and *OsHSFA3* overexpression transgenic *Arabidopsis* plants stressed for 20 days. (**B**) Survival rate of plants shown in (A) after rewatering. (**C**,**D**) Plant biomass and seedling fresh weight under control and drought stress for WT and *OsHSFA3*-overexpressing transgenic *Arabidopsis* plants. (**E**) Water loss rate of the detached leaves of *OsHSFA3*-overexpressing transgenic lines and WT. Values indicate mean ± SE of 10 biologically independent samples. Significance of data is tested with Student’s *t* test. ** represents *p* < 0.01.

**Figure 5 ijms-21-01857-f005:**
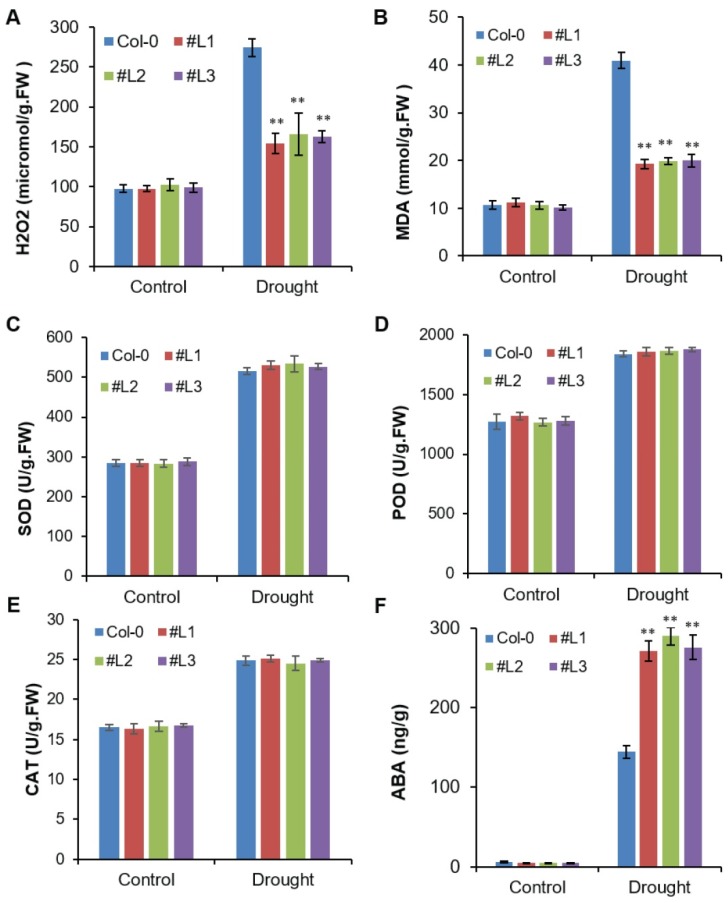
Reactive oxygen species (ROS) and antioxidant response under drought stress in *OsHSFA3*-overexpressing transgenic *Arabidopsis* plants. Quantitative measurement of H_2_O_2_ (**A**), malondialdehyde (**B**), superoxide dismutase (SOD) (**C**), peroxidase (POD) (**D**), catalase (CAT) (**E**) and ABA (**F**) in WT and *OsHSFA3*-overexpressing transgenic *Arabidopsis* plants under control and drought stress. Values indicate mean ± SE of 10 biologically independent samples. Significance of data is tested with Student’s *t* test. ** represents *p* < 0.01.

**Figure 6 ijms-21-01857-f006:**
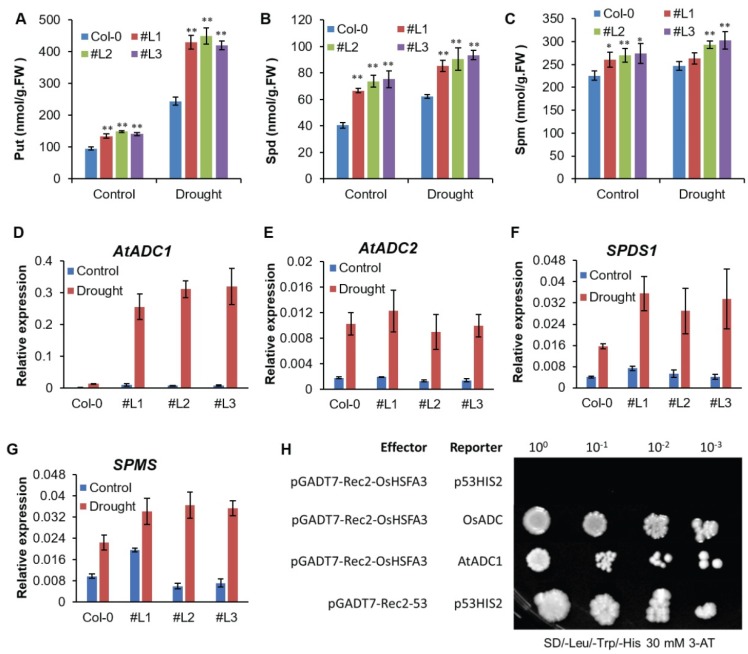
Analysis of polyamines, mRNA abundance and yeast one-hybrid assay for *OsHSFA3*. (**A**–**C**) Free polyamine contents in WT and *OsHSFA3-*overexpressing transgenic lines before and after drought stress. Put, putrescine; Spd, spermidine; Spm, spermine. Values indicate mean ± SE of 3 biologically independent samples. Significance of data was tested with Student’s *t* test. * represents *p* < 0.05, ** represents *p* < 0.01. (**D**–**G**) Quantitative real-time PCR analysis of expression levels of stress-responsive genes in WT and *OsHSFA3-*overexpressing transgenic plants under normal and drought conditions. (**H**) Yeast one-hybrid assay showing that *OsHSFA3* can bind to *AtADC1* and *OsADC* in vitro on a triple-dropout medium.

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
