# Peer review of "Rice *OsHSFA3* Gene Improves Drought Tolerance by Modulating Polyamine Biosynthesis Depending on Abscisic Acid and ROS Levels"

_ijms, 2020, doi:10.3390/ijms21051857_

Round 1

Reviewer 1 Report

Authors have depicted the potential role of Heat shock factors genes in rice drought tolerance. The manuscript is relatively well-performed. However, some flaws that can be corrected to improve the understanding the manuscript. Hence, recommend for major revision.

First of all in this manuscript lacking with heading and sub-heading section number. Besides, citation of reference are also not according to the journal format.

Line 165-168. Author have performed the tissue specific expression pattern of OsHSFA3. I think author should represent absolute expression of OsHSFA3, not their relative expression. In relative expression of q PCR, how author decide for comparison with other tissue and why?  (see the paper; Lu, Y et al., 2012 https://doi.org/10.1186/1746-4811-8-9).

Line 318: Arabidopsis thaliana should be italics. Please check the scientific name in reference section also and italics them.

Line 137 and 331: Putative HSE cis-element… and In silico Cis-element , cis should be italcis.

Lots of place space is missing, line 193, line 220,

Author did not presented the expression of empty vector containing (negative control) transformed Arabidopsis line which is essential for experimental procedure. Hence, expression of wild type, empty vector transgenic line and OsHSFA3 transgenic lines required for confirmation of gene function. Figure 4 must be inclusion of empty vector transformed line and their expression also must be depicted.

How much OsHSFA3 gene (used for transgenic line) gene identical from their orthologous Arabidopsis HSFA3 gene. Please point out it suitable section in manuscript.

Since rice is monocot and Arabidopsis is dicot. Hence, rice transgenic would be more reliable than Arabidopsis one. Why author choose Arabidopsis not rice for transgenic line. If Arabidopsis than figure 1(C and D) must be include Arabidopsis HSFA3 gene.

Author Response

Reviewer 1:

Authors have depicted the potential role of Heat shock factors genes in rice drought tolerance. The manuscript is relatively well-performed. However, some flaws that can be corrected to improve the understanding the manuscript. Hence, recommend for major revision.

Comment: First of all in this manuscript lacking with heading and sub-heading section number. Besides, citation of reference are also not according to the journal format.

Response: Thanks for your suggestion. We added the heading and sub-heading section number, and changed the reference format. Also, we change the structure of the manuscript along with the order: 1 Introduction, 2 Results, 3 Discussion, 4 Materials and Methods, 5 Conclusions; so that the manuscript accord with the journal format. And we submitted the manuscript with corrections tracks as a supplement file, we hope it is more convenient for you to revise our manuscript.

Comment: Line 165-168. Author have performed the tissue specific expression pattern of OsHSFA3. I think author should represent absolute expression of OsHSFA3, not their relative expression. In relative expression of q PCR, how author decide for comparison with other tissue and why?  (see the paper; Lu, Y et al., 2012 https://doi.org/10.1186/1746-4811-8-9).

Response: We used the actin gene as the reference gene, it supposed that actin gene was a housekeeping gene which would expressed consistently in all the tissues (we added this in M&M, line 332. There were many reports used the same method to conduct tissue specific expression pattern of genes (Int. J. Mol. Sci. 2018, 19(7), 2009, https://doi.org/10.3390/ijms19072009). So we thought it was a common method.

Comment: Line 318: Arabidopsis thaliana should be italics. Please check the scientific name in reference section also and italics them.

Response: Thanks for your suggestion. We checked the all and corrected them, and they were colored in red.

Comment: Line 137 and 331: Putative HSE cis-element… and In silico Cis-element , cis should be italcis.

Response: Thanks for your suggestion. We checked the all and corrected them, and they were colored in red.

Comment: Lots of place space is missing, line 193, line 220,

Response: Thanks for your suggestion. We checked the all and corrected them.

Comment: Author did not present the expression of empty vector containing (negative control) transformed Arabidopsis line which is essential for experimental procedure. Hence, expression of wild type, empty vector transgenic line and OsHSFA3 transgenic lines required for confirmation of gene function. Figure 4 must be inclusion of empty vector transformed line and their expression also must be depicted.

Response: Thanks for your suggestion, indeed it is more convincible to use empty vector transgenic line. However, pCAMBIA 1301 vector, were used in the transformation of Arabidopsis, which has been employed for transformation of in many experiments, and we have used it in many previous studies, so it might be unnecessary to develop and analyze empty vector transgenic lines here (Nguyen et al. 2019 (https://www.mdpi.com/2218-273X/9/11/714/htm); Jian et al. 2016 (https://journals.plos.org/plosone/article?id=10.1371/journal.pone.0149137); RAdi et al 2006 (http://www.plantphysiol.org/content/140/2/528); Uraguchi et al., 2019 (https://www.nature.com/articles/s41598-019-40671-x); Niu et al. 2018 (https://link.springer.com/article/10.1007/s00438-018-1452-3) etc. ).

Comment: How much OsHSFA3 gene (used for transgenic line) gene identical from their orthologous Arabidopsis HSFA3 gene. Please point out it suitable section in manuscript.

Response: Thanks for your suggestion. The homologous gene of OsHSFA3 in Arabidopsis thaliana (AT5G03720, AtHSFA3) is quite different in term of gene structure (two introns and three exons) and protein similarity (~ 40% identity), indicating that OsHSFA3 and AtHSFA3 may have distinct functional properties. According to your suggestion, we added this description in Results (line 73-75).

Comment: Since rice is monocot and Arabidopsis is dicot. Hence, rice transgenic would be more reliable than Arabidopsis one. Why author choose Arabidopsis not rice for transgenic line. If Arabidopsis than figure 1(C and D) must be include Arabidopsis HSFA3 gene.

Response: Thanks for your suggestion. Since there were low similarities of HSF3 (both gene structure and protein sequence) between Arabidopsis and rice as well as other transcription factors, so it's not appropriate to added it in Figure 1. Arabidopsis is a better model to study mechanism so we used the method of heterologous expression. But further we also genetic manipulation it and its homologous protein (many homologous proteins of transcription factor are redundancy, resulting in harder manipulation) in rice.

Reviewer 2 Report

 In this work, the authors overexpressed a rice Heat-Shock Factor gene (OsHSFA3), previously found to be strongly associated with drought tolerance, in Arabidopsis thaliana. To investigate the evolutionary relationship of OsHSFA3 with its homologs from different plant species, a phylogenetic tree was constructed using orthologs of OsHSFA3 proteins from 13 species. They examined the subcellular localization of OsHSFA3 protein and found it to be localized in the nucleus.

Phenotypic and functional assays of three transgenic Arabidopsis lines showed that over-expression of OsHSFA3 enhanced drought tolerance by reducing water loss and H2O2 production. Higher ABA and polyamine levels were observed in transgenic lines exposed to drought as compared to WT plants and this correlated with increased expression of polyamine biosynthetic genes, AtADC1, AtADC2, SPDS1 and SPMS. Via the yeast one-hybrid assay, they also show that OsHSFA3 binds to AtADC1 and OsADC.

The work is well planned and performed and the results are innovative and interesting.

Minor editing is required. In particular, the Abstract and Introduction require a throrough revision of the English grammar and syntax. Abbreviations should not be used in the Abstract.

Author Response

Reviewer 2:

Comments: In this work, the authors overexpressed a rice Heat-Shock Factor gene (OsHSFA3), previously found to be strongly associated with drought tolerance, in Arabidopsis thaliana. To investigate the evolutionary relationship of OsHSFA3 with its homologs from different plant species, a phylogenetic tree was constructed using orthologs of OsHSFA3 proteins from 13 species. They examined the subcellular localization of OsHSFA3 protein and found it to be localized in the nucleus.

Phenotypic and functional assays of three transgenic Arabidopsis lines showed that over-expression of OsHSFA3 enhanced drought tolerance by reducing water loss and H2O2 production. Higher ABA and polyamine levels were observed in transgenic lines exposed to drought as compared to WT plants and this correlated with increased expression of polyamine biosynthetic genes, AtADC1, AtADC2, SPDS1 and SPMS. Via the yeast one-hybrid assay, they also show that OsHSFA3 binds to AtADC1 and OsADC.

The work is well planned and performed and the results are innovative and interesting.

Minor editing is required. In particular, the Abstract and Introduction require a throrough revision of the English grammar and syntax. Abbreviations should not be used in the Abstract.

Response: Thanks for your comments and suggestions on our manuscript. And according to your suggestions, we modified the whole abstract, and the corrections were in tracked changes. And we submitted the manuscript with corrections tracks as a supplement file, we hope it is more convenient for you to revise our manuscript.

Round 2

Reviewer 1 Report

Revised manuscript have improved. Hence accepted in current form.